# DRPreter: Interpretable Anticancer Drug Response Prediction Using Knowledge-Guided Graph Neural Networks and Transformer

**DOI:** 10.3390/ijms232213919

**Published:** 2022-11-11

**Authors:** Jihye Shin, Yinhua Piao, Dongmin Bang, Sun Kim, Kyuri Jo

**Affiliations:** 1Interdisciplinary Program in Bioinformatics, Seoul National University, Seoul 08826, Korea; 2Department of Computer Science and Engineering, Institute of Engineering Research, Seoul National University, Seoul 08826, Korea; 3AIGENDRUG Co., Ltd., Seoul 08826, Korea; 4Interdisciplinary Program in Artificial Intelligence, Seoul National University, Seoul 08826, Korea; 5MOGAM Institute for Biomedical Research, Yongin-si 16924, Korea; 6Department of Computer Engineering, Chungbuk National University, Cheongju 28644, Korea

**Keywords:** transcriptomics, artificial intelligence, pharmacogenomics, human health, cancer, drug sensitivity, graph neural networks, Explainable AI, precision medicine, drug discovery

## Abstract

Some of the recent studies on drug sensitivity prediction have applied graph neural networks to leverage prior knowledge on the drug structure or gene network, and other studies have focused on the interpretability of the model to delineate the mechanism governing the drug response. However, it is crucial to make a prediction model that is both knowledge-guided and interpretable, so that the prediction accuracy is improved and practical use of the model can be enhanced. We propose an interpretable model called DRPreter (drug response predictor and interpreter) that predicts the anticancer drug response. DRPreter learns cell line and drug information with graph neural networks; the cell-line graph is further divided into multiple subgraphs with domain knowledge on biological pathways. A type-aware transformer in DRPreter helps detect relationships between pathways and a drug, highlighting important pathways that are involved in the drug response. Extensive experiments on the GDSC (Genomics of Drug Sensitivity and Cancer) dataset demonstrate that the proposed method outperforms state-of-the-art graph-based models for drug response prediction. In addition, DRPreter detected putative key genes and pathways for specific drug–cell-line pairs with supporting evidence in the literature, implying that our model can help interpret the mechanism of action of the drug.

## 1. Introduction

The advances in technology and scientific capability enable the acquisition of large amounts of personal omics data at a reduced cost [1]. Consequently, there is a growing interest in using individualized health data for precision medicine, leading to a number of data-driven healthcare models [2]. Pharmacogenomics, one of the branches of precision medicine, is the study of how a person’s genetic profile influences their response to medications [3,4]. Prediction of drug response or efficacy using the omics data of patients before the actual treatment is crucial because it can help increase clinical success and minimize adverse drug effects by modifying dosages or selecting alternative medications based on predicted value for personalized chemotherapy. However, obtaining patients’ tumor tissues via surgical procedure or biopsy involves safety issues [5], and performing animal experiments for clinical trials to infer human drug efficacy leads to ethical and financial concerns [6]. In addition, even though correlating the drug response and omics data can help improve understanding the drugs’ mechanisms of action [7], many candidate drugs still fail to enter clinical trials during the drug discovery process due to an incomplete understanding of the mechanisms [8,9]. In this respect, an interpretable in silico model for drug response prediction would be useful for numerous healthcare purposes, especially for precision medicine and drug discovery [10].

Molecular profiles of cancer cell lines and high-throughput drug sensitivity screening databases are publicly available [11,12,13,14,15,16], including CCLE (Cancer Cell Line Encyclopedia) [12] and GDSC (Genomics of Drug Sensitivity in Cancer) [13,16]. Public databases and improved computing technologies such as machine learning and deep learning have contributed to the rapid development of models for predicting anticancer drug sensitivity from cancer cell lines based on their genetic profiles.

The early studies in drug sensitivity prediction utilized machine-learning techniques [17,18,19] such as a random forest [20], support vector machine [21], and matrix factorization [22,23]. However, the traditional machine learning-based models can still be improved in terms of predictive performance and generalizability [3,24]. Matrix-factorization-based models leave nonlinear relationships unaddressed because they attempt to identify interactions between the drugs and cell lines using linear combinations of latent features. With the capabilities of learning complex nonlinear functions and high-dimensional representations from raw data, various deep learning techniques have been utilized for predicting drug responses [24], and the overall predictive power of drug sensitivity has been improved [25]. DeepDR [26] and MOLI [27] are drug-specific models that only use cell features such as somatic mutations, gene expression profile, and copy number variation to predict the IC50 values of each sample. tCNNs [28] introduced a model to predict drug sensitivity for drug–cell pairs using SMILES (Simplified Molecular Input Line Entry System) [29] sequences as drug features in addition to the genomic profiles of cells. The models described above used vector representations in common for describing cell or drug features.

Graph-based approaches have been introduced in drug-response prediction models to take advantage of the structural information of a drug or a gene network. A drug can be represented as a molecular graph consisting of a set of atoms (nodes) and a set of bonds (edges), and the graph is transformed into a high-level representation by a neural network [30,31]. For example, GraphDRP [31] obtained drug embeddings using graph convolutional networks, and cell line embeddings were derived from binary vectors of genomic aberrations. The state of a cell line can also be characterized as a gene–gene interaction network where genes (nodes) have node features from omics data such as gene expression values [32,33,34]. Reference [33] proposed an end-to-end drug-response prediction model, TGDRP, with cell-line graph embedding consisting of genes that have cancer-related mutations and drug graph embedding obtained by a graph neural network. They also proposed TGSA, which updates embeddings from TGDRP with similarity information between cell lines and drugs and predicts the drug response again. Among graph-based drug-response prediction models, NIHGCN [35] constructed a cell-line–drug heterogeneous network with cell-line gene expression and drug-fingerprint vectors as node features to aggregate neighborhood interactions of drug and cell line. Then, there are two different types of GCN layer for aggregating both homogeneous and heterogeneous neighbors’ information for drug response prediction. However, even though it is a graph-based model, the biochemical structures of cell line and drug are not reflected in the model.

While the recent studies described above have introduced graphs into the deep-learning models to leverage structural information and improve prediction accuracy, the models lack interpretability of the predicted results. Several methods tried to delineate the mechanisms governing the drug responses, highlighting the important genes or high-level subsystems, such as biological pathways that can cause changes in cellular phenotype. SWnet [36] explored the interactions between genetic profile and the chemical structure of drugs using self-attention and identified genes with the strongest predictive power. Reference [37] proposed a multi-layer perceptron model called pathDNN which incorporates a layer of pathway nodes and quantified the activity of each pathway to explain its effect on the drug response. DrugCell [9] obtained binary encodings of mutational status via a visible neural network guided by a hierarchy of cell subsystems and measured the predictive performance of the subsystems. Although pathDNN and DrugCell attempted to construct an explainable model with a hierarchical structure, biological pathways were implemented as gene sets rather than gene networks, indicating that domain knowledge in gene–gene interactions is not fully reflected in the models.

According to the existing studies that suggest deep-learning models for drug response prediction, it is helpful to incorporate graph representation for both drug and cell line profiles, which enables detailed descriptions of compound structures and gene networks. Moreover, the gene network can be dissected as a set of biological pathways that include gene–gene interactions for each specific biological mechanism, which can help enhance both prediction accuracy and interpretability. However, current interpretable models for drug response prediction simply describe the network as gene-pathway layers, leaving the interaction information inside the biological pathways unused. Here, we propose a novel anticancer drug-response prediction model named DRPreter (drug response predictor and interpreter) with key features as follows:*Knowledge-guided cell representation with graphs.* DRPreter constructs a cell line network as a set of subgraphs that correspond to cancer-related pathways for the detailed representation of the biological mechanism.*Interpretability of drug mechanisms of action.* Using the transformer’s encoder, the interactions between drugs and pathways are derived from the model, and putative key pathways for the drug mechanism can be highlighted.*Enhanced performance.* DRPreter outperforms state-of-the-art drug-response prediction models, as demonstrated by comparative experiments on the GDSC drug-sensitivity dataset.

## 2. Results and Discussion

In this study, we developed a regression model to predict the half maximal inhibitory concentration (IC50), normalized to natural logarithms, which is a representative indicator of drug sensitivity. The following is a description of the graph configuration for cell lines and drugs and the graphical abstract of DRPreter (Figure 1).

The main steps of DRPreter are: cell-line graphical representation, drug graphical representation, and drug response prediction module. As a first step in creating the cell-line graph representation, a template graph was created with all genes involved in the selected 34 cancer pathways as nodes and gene–gene interactions between the genes as edges. After that, each cancer pathway was used as a predefined subgraph of the template graph, and each pathway embedding was derived using graph attention networks (GAT). As a next step in creating drug-graph representation, we transformed SMILES-format drug structures into graphs with atoms as nodes and bonds as edges, and used a graph isomorphism network (GIN) as a graph encoder to generate drug embeddings. With the 34 pathway embeddings and drug embeddings obtained as inputs to the transformer-based cell line and drug fusion module, the embeddings were updated by reflecting inter-pathway relationships and pathway-drug relationships in the model learning process. Updated pathway embeddings were combined into graph-level embeddings for the entire cell line through concatenation, and graph-level drug embedding was obtained by combining embeddings before and after passing through the transformer encoder. In conjunction with two graph level embeddings, the IC50 of a given cell-line–drug pair was predicted using a multi-layer perceptron.

### 2.1. Performance Comparison

#### 2.1.1. Dataset

In the cell line template graph, the initial feature of each gene node was derived from transcriptomic data of each cell line obtained from the CCLE database version of 21Q4 [38] (https://portals.broadinstitute.org/ccle, accessed on 3 December 2021). The gene expression data were TPM values of the protein-coding genes for DepMap cell lines, which were inferred from RNA-seq data using the RSEM tool and were provided after log2 transformation, using a pseudo-count of 1; $log_{2}$(TPM+1) [38]. We assigned edges of the graph as only those interactions with high reliability scores and a combined score of at least 990 among the STRING (v11.5) [39] protein–protein interactions. The edges of the template graph and each subgraph were all STRING protein–protein interactions. Only the genes corresponding to each cancer-related pathway were obtained in KEGG, and the genes corresponding to each pathway were used as nodes in the subgraph. The STRING interactions were used as the edges connecting them. Pathways for constructing subgraphs were selected in the following manner. The non-processed pathways listed in categories 6.1 and 6.2 of the KEGG pathway database (https://www.genome.jp/kegg/pathway.html, accessed on 16 April 2022) were categorized according to the cancer types. These pathways include common subpathways related to cell signaling, the cell cycle, and apoptosis, which are key in various types of cancer. Consequently, if the cancer pathways provided by KEGG are used as they are, the overlap between the pathways will be excessive, and the meaning of learning for each pathway diminishes. Additionally, KEGG provides information on detailed pathways associated with each cancer type pathway. There were a total of 84 detailed pathways categorized by function. Among these pathways, we eliminated duplicate pathways, metabolic pathways, non-cancer disease pathways, viral infection pathways, and pathways with fewer than 10 genes or gene–gene interaction edges. Additionally, the focal adhesion pathway (hsa04510) was also eliminated because 91% of the genes constituting this pathway were included in the remaining pathways. The finally selected 34 cancer-related detailed pathway list can be found in Table A1. For drug graph construction, we obtained SMILES strings from PubChem [40].

For the performance comparison experiment, we compared our method with state-of-the-art GNN-based drug-response prediction models obtaining either cell-line or drug embedding using a homogenous biochemical structure-based graph: GraphDRP, TGDRP, and TGSA. As the initial feature for each gene node, GraphDRP uses mutation (mut) and copy number variation (cnv); and TGDRP and TGSA use mut, cnv, and gene expression (exp). As GraphDRP represents cell lines as one-dimensional binary vectors, a one-dimensional CNN is used to get their embeddings. Cell lines and drugs are represented in graph format in TGDRP and TGSA, and the embeddings are obtained by an GNN. The cancer driver genes from COSMIC were selected as the genes to represent the cell lines in all baseline models [41]. The COSMIC database provides information about mutation-containing genes involved with cancer, and how these mutations can cause cancer. We selected 702 COSMIC Cancer Gene Census (https://cancer.sanger.ac.uk/cosmic/census?tier=all, accessed on 3 December 2021) genes; all three omics data types are provided in CCLE. We used the genes equally for the baseline model execution. Moreover, the types of cell lines and drugs used in this study were the same as in the TGDRP and TGSA. The data type used by our model differed from that of every baseline model, and ours used the most numerous omics types among them. To use only cell-line–drug pairs with three types of omics data available, intensive filtering was done on cell lines and drugs. Since all omics data had to be imported for baseline model execution, the same cell-line–drug pair was used as in the most data-intensive models. Consequently, the performance test consisted of 580 cancer cell lines that can obtain omics data from CCLE and 170 anticancer drugs provided by GDSC2. The total number of possible cell-line–drug pairs was 82,833 with log-normalized IC50 values.

In addition, as performance comparisons with deep learning, we added random forest and support vector machine (SVM) as baseline models for comparative analysis. For a fair comparison, we used the same features and preprocessed the data to feed into traditional machine-learning methods. For cell line embedding, we concatenated all gene expression vectors, resulting in a one-dimensional vector, and for drug embedding, since different drugs have different atoms, we simply the sum of each atom embedding, resulting in a one-dimensional vector. Finally, we concatenated cell line embedding and drug embedding to one-dimensional embedding and fed it into the models. The nodes constituting the cell-line graphs of the existing GNN-based drug-response prediction models were configured according to the settings in each comparison paper.

#### 2.1.2. Experimental Setups

In the regression experiments for predicting natural log-transformed IC50 values based on drug and cancer cell line profiles, we used four standard evaluation metrics to compare the results of different models by computing the statistical correlation and accuracy of predicted and observed IC50 values. The metrics included the Pearson correlation coefficient (PCC), Spearman correlation coefficient (SCC), root absolute error (MAE), and mean-squared error (MSE). PCC measures the linear correlation of observed and predicted IC50 values. SCC is a non-parametric measure for rank correlation between observed and predicted IC50 values. MSE and MAE directly measure the difference between observed and predicted IC50 values.

#### 2.1.3. Rediscovered Responses of Known Pairs

All possible cell-line–drug pairs were randomly divided into training, validation, and test datasets in an 8:1:1 ratio, and the experiments were conducted repeatedly on 10 random seeds. For each model, the test performance was averaged over the seeds and is reported as mean ± standard deviation. Comparing the results of different models was based on four common evaluation indicators. The mean-squared error and mean absolute error between the predicted IC50 and the true IC50; and the Pearson correlation coefficient and Spearman correlation coefficient of each IC50 distribution were used as evaluation criteria. Compared to the baseline model we selected above, we conducted an ablation study to examine each part of DRPreter’s effectiveness (Table 1). Based on the results of the ablation study, we assume that network information from biological pathways helped improve the performance of our model. It has also been found that the ability to relate the cell line in its pre-drug treatment state to the drug through the transformer has a significant effect on performance improvement. In addition, DRPreter showed a performance improvement of about 20% in MSE compared with the next best model (Table 2). We also conducted internal validation using a 10-fold cross-validation experiment on the original dataset using CCLE and GDSC2. The validation set was selected by five random seeds (Table 3).

### 2.2. Case Study

#### 2.2.1. Interpolation of Unknown Values

The method of missing values prediction has been widely used in drug-response-prediction studies [28,30,31,33] to identify whether the model is capable of inductive prediction. For evaluating the inductive predictability of our model, we trained with all the known cell-line–drug pairs and predicted values without experimental results of pairs in the GDSC2 database. There were a total of 98,600 pairs using 580 cancer cell lines and 170 drugs, but 15,767 cell lines were not covered by our data due to filtering because of a lack of omics data or due to the absence of drug response experiments in GDSC. The model with the highest performance was used to predict missing drug response values.

We illustrate the distributions of known IC50 values in GDSC2 and the predicted values of our model (Figure 2). The box plots are grouped by drugs, and each box represents the distribution of the IC50 values within a cell line. We displayed the drugs with the top 10 highest and top 10 lowest median IC50 value. After conducting Mann–Whitney Wilcoxon test for each drug distribution, 18 drugs among the 20 selected drugs showed no significant difference between the GDSC2 and predicted unknown IC50 value distribution. The result implies the predicted missing IC50 values follow the measured value distribution.

The total predicted missing values using our model can be found in Appendix A of the Appendix A.

Not knowing the actual values for these missing pairs, we conducted literature searches to assess our predictions. Bortezomib had the smallest overall IC50 distribution, and the most sensitive cell-line pair was LP-1 in our model. LP-1 is a cell line derived from the peripheral blood of a multiple myeloma patient. Bortezomib is a proteasome inhibitor that is widely used in patients with multiple myeloma [42,43]. Rapamycin was not included among the top 10 sensitive drugs in the known GDSC data but in our predicted values, so we analyzed it further. In our study, rapamycin was most sensitive to the MV-4-11 cell line. The MV-4-11 cells are macrophages that were isolated from the blast cells of a biphenotypic B myelomonocytic leukemia patient. Rapamycin can inhibit leukemic activity in acute myeloid leukemia by mTOR inhibition through the blockade in G0/G1 phase of the cell cycle [44].

Based on the biological processes at the cellular and molecular level of cancer cells and drugs, DRPreter can make inductive predictions for cell lines and drugs when there are no known responses and seems to have the potential to select candidates for drug treatment.

#### 2.2.2. Gradient-Weighted Gene Nodes Interpretation

It is essential for drug-response prediction methods to capture significant biological implications and to make accurate predictions. A gene-level analysis was performed first to determine whether the model was taking into account genes that are known as drug targets, involved in target pathways, or biomarkers of disease. We prioritized genes from an input drug and cancer–cell-line pair by scoring each gene with a gradient-weighted extent to check whether it is drug-target-related. The importance score of each gene node was determined by GradCAM, which is a widely utilized technique to produce explanations of model decisions [45], and we considered the score as the extent of its contribution. In our model, GradCAM determined the influence of input gene nodes on the label by tracing back the gradient backpropagation process of the model for predicting IC50 value. Table 4 shows the top five most significant genes of each cell-line–drug pair in the test dataset.

As verified by literature searches, the bold genes in Table 4 are the target genes or genes associated with the target pathway for each drug–cell-line pair. The targets were obtained from DrugBank [46] and GDSC, and the genes corresponding to the target pathway were obtained from GeneCards [47] and Harmonizome [48]. Afatinib is a irreversible ErbB family blocker [49] that targets *EGFR* and *ERBB2*, and its target pathway is the EGFR signaling pathway. Our model found *ERBB2* as a significant gene of the afatinib pair. Among the other top five genes, *LTBR* was found to be related with tumor treatment for its potential in triggering apoptosis of tumor cells or anti-tumor immune response [50]. As with the majority of cancers, *TP53* is the most common mutated gene, showing a predominant clonal expression in Non-Small-Cell Lung Cancer (NSCLC) [51]. Additionally, microtubule-active drugs, including vinblastine, are known to induce apoptosis through inducing expression of p53 protein [52]. It is known to be possible to use *CLDN18* as an early-stage indicator of pancreatic ductal carcinogenesis and to study *CLDN18*’s regulatory mechanisms for uncovering key pathways such as the PKC pathway of pancreatic cancer [53]. *WNT7A*, the fourth-ranked gene, shows relation with docetaxel for Wnt signaling, playing a role in docetaxel resistance [54]. *CDK2* corresponds to the mTOR signaling pathway, which is the target pathway of rapamycin. Additionally, an in vivo experiment reported that upregulation of TYRP1 and TYR proteins may explain the melanogenesis of rapamycin-treated melanoma cells [55]. The use of bortezomib and paclitaxel suggests the potential for rationally designed treatments for solid tumors with MAPK pathway activation [56].

#### 2.2.3. Pathway-Level Interpretation Using the Transformer

We examined which pathways were stimulated in various cancer types that are sensitive to drugs and some that are not, and whether our model could capture such meaningful pathways. The self-attention score from the transformer-based structure (Figure 3) was investigated for a drug that is sensitive only to specific cell lines. All the GDSC data with known IC50 values were observed in the same way as Figure 2a, and dasatinib was selected as having the widest IC50 distributions. The wide distribution of its IC50 means that the drug exhibits the greatest differences in efficacy based on the type of cell line. We compared the self-attention score of the transformer on MEG-01, the cell line judged to be sensitive by having the smallest IC50 value, and BT-483, the most insensitive cell line with the largest IC50 value, among the 548 cell line pairs with dasatinib (Figure 4). The MEG-01 cell line was derived from the hematopoietic and lymphoid tissue of a leukemia patient, and the BT-483 cell line was derived from the breast tissue of a breast-cancer patient.

The TGF-𝛽 signaling pathway (hsa04350) was the pathway with the highest attention score for the MEG-01 cell line, which is the most sensitive to dasatinib. The second-most-important pathway, ubiquitin-mediated proteolysis (hsa04120), involves the covalent binding of ubiquitin to the target protein and its degradation. It is known that ubiquitin-mediated degradation can regulate the TGF-𝛽 signaling pathway [57]. The TGF-𝛽 signaling pathway suppresses tumors in normal and premalignant cells, yet promotes oncogenesis in advanced cancer cells, and its components are regulated by ubiquitin-modifying enzymes; abnormalities of the enzymes can cause malfunctioning of the pathway, which can cause cancer, tissue fibrosis, and metastasis [58,59,60]. In this regard, the ubiquitin-modifying enzymes in the pathway and their counterparts are increasingly being explored as potential drug targets [59]. Dasatinib is a tyrosine kinase inhibitor that can be a treatment for chronic myeloid leukemia [61]. Dasatinib functions by binding to the ATP site of the active conformation of BCR-Abl [62]. As a signal-transduction inhibitor, dasatinib inhibits the proliferation of tumor cells by inhibiting tyrosine kinase action, especially blocking transcriptional and promigratory responses to TGF-𝛽 through inhibition of Smad signaling [63]. The ubiquitin pathway can regulate the basal level of Smads, and altered Smad proteins can cause a malfunction in responding to the incoming signals due to their importance in transducing TGF-𝛽 signals [57]. From the ubiquitin to the TGF-𝛽 pathway, our model captures the mechanism of action of the drug.

Moreover, the ECM–receptor interaction pathway (hsa04512) was found to be most important in the breast-cancer cell line, which is the most insensitive to dasatinib. The ECM–receptor interaction pathway has been shown to be possibly useful as a biomarker for breast cancer [64], but it does not relate to dasatinib’s mechanism of action. Hence, our model identified the pathways related to the drug’s mechanism of action for drug-sensitive carcinoma and focused on the biomarker for carcinoma without drug efficacy.

## 3. Materials and Methods

### 3.1. Graph Neural Networks

A graph neural network (GNN) is a type of neural network that operates on graph-structured data. GNN uses the topology of the graph to learn the relationships between the input features. It can perform more effectively than other representation learning methods on input data with topological information. In this study, we represent a graph as *G* = (*V*, *E*) where $V=\{v_{1},\ldots ,v_{n}\}$ is the set of *n* nodes and E⊆V×V is the set of edges. The node $v_{i}$ has node feature $x_{i}\in$$R^{d}$, where *d* is a dimension of the feature. The node feature matrix of the graph can be represented as X∈$R^{n\times d}$, where *n* is the number of nodes in the graph. Adjacency matrix A∈$R^{n\times n}$ indicates the total connectivity of nodes in the graph, where $A_{i,j}=1$ means nodes, $v_{i}$ and $v_{j}$ are linked, and $W^{\left(l\right)}$ represents the parameters of the *l*-th layer of the graph (Table 5).

In each GNN layer, a key mechanism, called message passing, updates the node representation by using the node features of the previous layer and the topology of the graph [65]. The message passing mechanism involves aggregating the information of neighboring nodes and updating the hidden state of each node by combining the node representation from the previous layer and the aggregated messages. For every node in each layer, a transformed feature vector is generated capturing the structural information of the k-hop neighbor nodes. The GNN can update the *i*-th node representation in the *l*-th layer as in the following Equation [66,67], where *N*(*i*) is the set of neighbor nodes linked to the target node *i*. For a given node, the AGGREGATE step applies a permutation invariant function to its neighboring nodes to produce the aggregated node feature of neighbors, and the COMBINE step delivers the aggregated node feature to the learnable layer to produce updated node embedding by integrating the existing embedding and the aggregated neighbor embedding.
(1)$$x_{i}^{\left(l\right)}=COMBINE^{\left(l\right)}\left(x_{i}^{\left(l-1\right)},AGGREGATE^{\left(l-1\right)}\left(x_{j}^{\left(l-1\right)}:j\in N(i)\right)\right)$$

### 3.2. Cell-Line Graph Representation

#### 3.2.1. Cell-Line Graph Construction

We used a biological template network to represent cell lines to simulate gene–gene interactions in actual cells. In the cell-line graph $G_{c}$, genes are represented by nodes and edges represent the relationships between genes. This template graph contained 2369 genes selected using the pathway selection method described in the next section.

It is known that drugs do not have a universal effect throughout all cellular components, but tend to have distinct effects on specific genes or pathway targets. In this way, cancer cells undergo phenotypic changes as a result of drug molecules inhibiting or activating their target pathways. Motivated by this point, instead of representing the cell line as a homogeneous large-scale graph that contains the entire genes, we divided the template network $G_{c}$ into pathway subgraphs $G_{p}$ according to the biological domain knowledge inspired by [68] and learned graph embeddings from the selected subgraph units. Finally, the divided cell-line graph $G_{c}^{{}^{\prime }}$ was represented as a heterogeneous graph containing multiple subgraphs. We selected pathways that can be targeted by drugs, as they are associated with cancer from the KEGG pathway database [69], and used these pathways as pre-defined subgraphs of the cell-line template.

In the case of template graph $G_{c}$, the *i*-th selected pathway subgraph can be described as $G_{p}^{\left(i\right)}$ = ($V_{p}^{\left(i\right)}$, $E_{p}^{\left(i\right)}$), where $V_{p}^{\left(i\right)}$ refers to a set of nodes and $E_{p}^{\left(i\right)}$ refers to a set of edges of the pathway. Thus, the template graph $G_{c}$ is extended as a union of disjoint graphs, with overlaps between the pathways in the form of $G_{c}^{{}^{\prime }}$ = {$G_{p}^{\left(1\right)}$, …, $G_{p}^{\left(34\right)}$}. In the template cell-line graph $G_{c}$, gene sets included in 34 pathways were represented by 2369 nodes and 7954 edges. A divided template graph $G_{c}^{{}^{\prime }}$ with pathways as subgraphs had 4646 nodes and 12,004 edges after combining the data from all pathways. The types of constituting genes remained the same, but the numbers of nodes and edges increased when the template network was divided into subnetworks due to the overlap of functions.

#### 3.2.2. Cell-Line Graph Encoder on Pathway Subgraphs

Transcriptomic features of nodes and biological network topology were captured within each subgraph using a graph attention network (GAT) [70]. Using the self-attention mechanism, GAT calculates a normalized attention score $\alpha_{ij}$, indicating the importance of the features of the neighbor nodes for a target node *i*, where j∈N(i). A subsequent step in the message passing process is for each node to reflect the importance of the neighboring nodes’ information in accordance with the previously obtained attention scores. Details and graphical overview of GAT can be found in the original paper [70].
(2)$$X^{\left(l\right)}=\sigma \left(\Sigma_{j\in N(i)}\alpha_{ij}^{\left(l-1\right)}W^{\left(l-1\right)}X^{\left(l-1\right)}\right)$$

If template graph $G_{c}$ is used as it is, edges connected to one gene include interactions from multi pathways, which can be noise. Node representations were updated through GAT on the cell-line graph constructed in the previous subsection. The cell-line graph consists of pathway-based subgraphs; thus, the updated node representation can reflect the intra-pathway gene–gene interaction information. To pool the cell-line graph-level embedding, we initially used simple hierarchical permutation-invariant graph-pooling strategies [71,72,73]. However, the graph-pooling strategies we employed resulted in slight performance degradation. We assumed that this was due to the relatively large size of the cell-line graph, and simply pooling the nodes into a vector of the same dimension may lose the information of the nodes in a cell line. As a result, the embeddings of each node learned through GAT were concatenated to form a graph-level embedding for each pathway.

### 3.3. Drug Graph Representation

#### 3.3.1. Drug Graph Construction

We used a graph neural network to learn the drug representation by reflecting the relationships between atoms connected by bonds and the overall molecular structural information. A drug can be represented as a graph in which atoms are nodes and bonds are edges. We used RDKit [74] to transform SMILES [29], a one-dimensional string format drug structure, into a graph format that can reflect structural information of an actual drug. The ten initial features of atomic nodes were imported from previous research [30,33], which predicted drug sensitivity from GNN-based embeddings of drug structures. The details of atomic and bond features can be found in Table A2.

#### 3.3.2. Drug Graph Encoder

We used the graph isomorphism network (GIN) [75] to learn the features of the atomic nodes within the drug graph. GIN applies a neighborhood aggregation method similar to the Weisfeiler–Lehman test [76] and updates the *i*-th node feature of the *l*-th layer as follows.
(3)$$x_{i}^{\left(l\right)}=MLP^{\left(l\right)}\left(\left(1+\epsilon^{\left(l\right)}\right)\cdot x_{i}^{\left(l-1\right)}+\Sigma_{j\in N(i)}x_{j}^{\left(l-1\right)}\right)$$

Details and graphical overview of GIN can be found in the original paper [75]. The graph encoder was chosen following the results of GraphDRP, which involved a comparison of different types of graph neural networks—GIN, GAT, and GCN+GAT—in order to analyze the effectiveness of each graph encoder in predicting drug response. In addition, GIN was widely used for the embeddings of drug graphs in various drug-response prediction models [31,33,34,77]. All embeddings of each atom node were updated through GIN, then they were concatenated to create raw drug embeddings before the pathway affected them.

### 3.4. Drug Response Prediction Module

#### 3.4.1. Knowledge-Guided Cell-Line–Drug Fusion Module Using Transformer

The transformer model tracks relationships in sequential data, such as the words in a sentence, to discover context and meaning from the components [78]. We used a transformer-based module to reflect not only inter-pathway interactions but also the interactions between the pathways and each drug, which would allow exploring the pharmacological mechanisms of action at the pathway level during a therapeutic process (Figure 3).

The embedding of 34 pathways and that of a drug obtained from graph representation modules were updated in the transformer-based cell line and drug fusion module. The encoder is based on the transformer encoder, and does not use positional encoding, as the order of the embeddings does not matter. Instead, we conducted two experiments: first with binary-data-type tokens indicating whether the embedding is a pathway or a drug, and then without positional encoding or data-type tokens. All-pairwise self-attention scores can be obtained between pathways and drug embeddings via multi-head attention. As a result of the encoder-based transformer model, the raw pathway embeddings were updated to drug-aware pathway embeddings according to the effects of the drugs, and the raw drug embedding was updated based on the relationship with each pathway.

Our structure has a single encoder-based layer taking pathway embeddings $X_{p}^{\left(l\right)}$ (*l* = 1, …, 34) and a drug embedding $X_{d}$ derived from knowledge-guided GNNs as input values. Inputs in a typical transformer’s encoder are constructed by adding positional encoding to embeddings of source sequences. Unlike translation, where an order of words in a sentence is important, the pathway embeddings entering our encoder are not affected by the order in which they are encoded, so a transformer’s encoder structure other than positional encoding was used for this study. As an alternative, we added a type-encoded token that indicates whether the embedding is a drug or a pathway. In an element-wise manner, type-encoded binary tokens are added to the input feature matrix with the same number of dimensions before input embeddings are fed into the module.

On the fusion pathway and drug embeddings, self-attention was performed several times through multi-head attention, and the average of each trial was used as the final attention score. After the encoder completed its execution, the encoder produced drug-aware updated pathway embeddings $X_{p}^{\left(l\right)}$’ and pathways’ transcriptome-aware updated drug embedding $X_{d}$’ reflecting interaction information. These drug-aware pathway embeddings facilitate interpreting the medication’s mechanism of action, since they can reflect both the drug–pathway interaction information and the interactions between the pathways. Drugs have a large structural variation when compared to cell-line graphs which are composed of the same genes and are structurally equal but have different node feature values. Therefore, it is possible that the variation of the drug embedding may be blurred because the new drug embedding updated as a result of the transformer is affected by the cell line embedding. Hence, we connect the raw drug embedding obtained through GNN prior to the transformer with the updated drug embedding obtained after the transformer using residual connection [79]. By residual connection, it is possible to preserve the original drug structure information and utilize the cell-line–drug interaction information using the updated drug embedding which recognizes the transcriptomic information of each pathway. We concatenated the resulting 34 subgraph embeddings in order to prevent information loss, thereby embedding the entire cell line.

#### 3.4.2. Improving Predictive Performance Using a Similarity Graph

Based on the idea that similar drugs and similar cell lines exhibit interchangeable drug response behaviors, some drug-response prediction models use prior knowledge of drug and cell-line similarity to minimize differences between drugs and cell lines in the latent space. Reference [22] applied regularization terms based on chemical-structure similarities between drugs and similarities between cell lines based on gene expression profiles to improve prediction accuracy and prevent overfitting.

We followed the similarity-based embedding updating strategy of [33]. From the completed end-to-end model up to Section 3.3, embeddings of all 580 cell lines and 170 drugs were created. Then, we constructed two homogeneous graphs, each consisting of cell lines and drug nodes, with the initial feature of each node set having the resulting embeddings of the previous step. Using GraphSAGE [80], we updated the embeddings of each homogeneous cell line and drug graph. After that, we updated embeddings of each cell-line–drug pair from two homogeneous graphs. We concatenated two embeddings into the one-dimensional vector and used a multi-layer perceptron to predict final IC50 values.

## 4. Conclusions

In this paper, we proposed an interpretable drug-response prediction model called DRPreter which integrates biological and chemical-domain knowledge with cutting-edge deep learning technologies to deliver outstanding predictive performance and interpretability. We introduced cancer-related pathways and constructed the cell line network as a set of subgraphs to represent and interpret biological mechanisms in detail. We extracted drug–pathway interaction information from the modified encoder of the transformer module and obtained putative key pathways for the drug’s mechanism. Ablation studies verified the effectiveness of each component of the model, and performance comparison experiments showed DRPreter has enhanced predictive power compared to the state-of-the-art graph-based drug-response prediction models which obtain either the cell line or drug embedding using a homogeneous biochemical structure-based graph.

To properly apply the drug response predicted by the model for clinical use or drug discovery, it is essential to understand the process and mechanism from which it was derived due to safety and reliability issues. Accordingly, we implemented gene and pathway-level analysis via DRPreter, and it has been shown that DRPreter predicts drug sensitivity based on known drug mechanisms of action and target-related factors. We also identified the cell line that would act most sensitively for each drug in the absence of experimented data through a case study and confirmed that it is widely used for each drug currently in the clinical situation. By doing so, patients who have shown resistance to a specific drug may be able to select a drug candidate group that would replace the unsuitable drug. It will be remarkably efficient to have comprehensive public databases of drug targets and predictive models that can interpret pharmacological mechanisms for personalized medicine and drug discovery. 

## Figures and Tables

**Figure 1 ijms-23-13919-f001:**
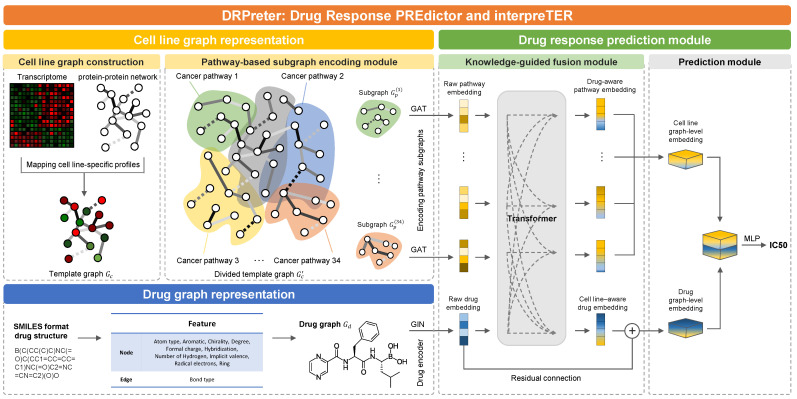
An overview of DRPreter. In the graph representation sections, embeddings of pathway subgraphs and drug molecule were obtained using GNN. With the obtained pathway embeddings and drug embeddings as inputs to the transformer-based cell-line and drug fusion module, the embeddings were updated by reflecting inter-pathway relationships and pathway-drug relationships in the model learning process.

**Figure 2 ijms-23-13919-f002:**
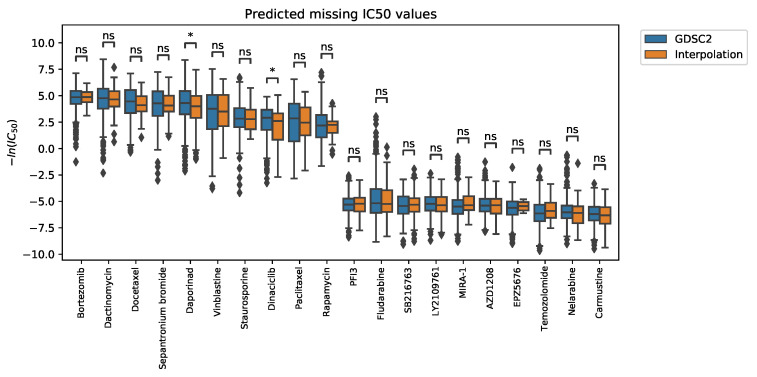
Box plot of drug-specific IC50 distributions of cell lines. The distribution of GDSC2 data (blue) compared with predicted missing IC50 values (orange). The 10 drugs with the highest median IC50 values and the 10 drugs with the lowest median were selected. Among the 20 drugs, IC50 value distributions of 18 drugs showed no significant differences through the Mann–Whitney Wilcoxon Test. ns: not significant, *: 0.01 < *p*-value < 0.05.

**Figure 3 ijms-23-13919-f003:**
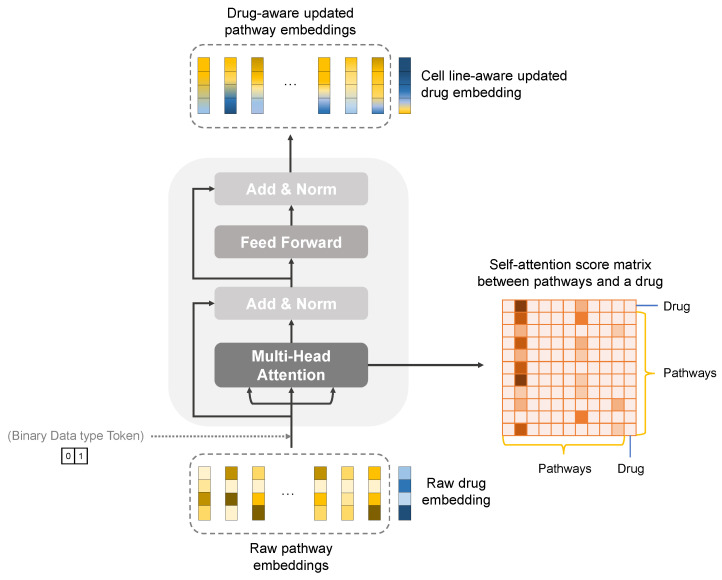
A detailed structure of type-aware transformer encoder reflecting interactions and relationships between pathways and a drug. We extracted drug-pathway interaction information from the modified encoder of the Transformer module and identified putative key pathways for the drug’s mechanism of action using a matrix of self-attention scores between pathways and the drug.

**Figure 4 ijms-23-13919-f004:**
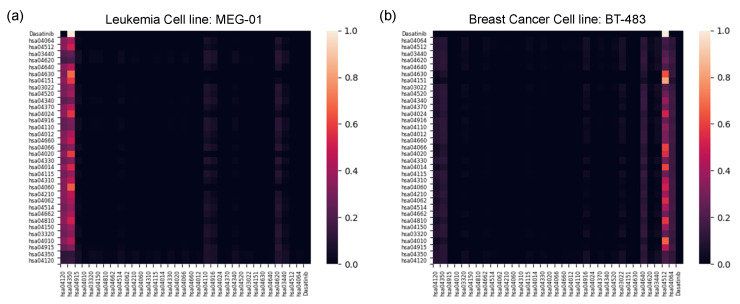
Visualization of all-pairwise self-attention scores from the transformer. (**a**) Dasatinib and leukemia cell line MEG-01 pair. (**b**) Dasatinib and breast cancer cell line BT-483. The figures show the y-axis as the query of the transformer and the x-axis as the key. On each axis, there is a drug and 34 pathways which start with “hsa”, indicating KEGG pathway identifiers.

**Table 1 ijms-23-13919-t001:** Model ablation studies with different settings.

Structural Settings of DRPreter	Data	MSE (↓)	MAE (↓)	PCC (↑)	SCC (↑)
Template graph	COSMIC ^1^	0.8926 ± 0.0363	0.6909 ± 0.0146	0.9423 ± 0.0027	0.9196 ± 0.0034
Template graph	Pathway ^2^	0.8536 ± 0.0420	0.6759 ± 0.0161	0.9449 ± 0.0032	0.9224 ± 0.0035
Pathway	Pathway ^2^	0.8645 ± 0.0277	0.6791 ± 0.0113	0.9446 ± 0.0014	0.9233 ± 0.0008
Pathway + Transformer	Pathway ^2^	0.8302 ± 0.0156	**0.6676** ± **0.0051**	0.9465 ± 0.0015	0.9242 ± 0.0015
Pathway + Transformer + Similarity	Pathway ^2^	**0.8251** ± **0.0122**	0.6682 ± 0.0047	**0.9467** ± **0.0013**	**0.9248** ± **0.0014**

^1^ COSMIC: 702 COSMIC genes. ^2^ Pathway: 2369 genes of 34 cancer-related pathways. The best performance is
shown in bold for each metric.

**Table 2 ijms-23-13919-t002:** Performance comparison with baseline models.

Model	Cell Encoder	Data	MSE (↓)	MAE (↓)	PCC (↑)	SCC (↑)
SVM ^1^	-	Pathway	8.5780 ± 2.0615	2.2976 ± 0.3005	0.5282 ± 0.0355	0.4471 ± 0.0476
RF ^2^	-	Pathway	1.6711 ± 0.0422	0.9608 ± 0.0100	0.8887 ± 0.0021	0.8497 ± 0.0034
GraphDRP	1D CNN	COSMIC	1.0110 ± 0.0157	0.7618 ± 0.0083	0.9386 ± 0.0018	0.9151 ± 0.0021
TGDRP	GNN	COSMIC	0.9004 ± 0.0341	0.6933 ± 0.0148	0.9417 ± 0.0026	0.9188 ± 0.0040
TGSA	GNN	COSMIC	0.8955 ± 0.0536	0.6913 ± 0.0238	0.9425 ± 0.0043	0.9201 ± 0.0051
DRPreter	Knowledge-guided GNN	Pathway	**0.8251**±**0.0122**	**0.6682**±**0.0047**	**0.9467**±**0.0013**	**0.9248**±**0.0014**

^1^ SVM: support vector machine. ^2^ RF: random forest. The best performance is shown in bold for each metric.

**Table 3 ijms-23-13919-t003:** Internal validation using 10-fold cross-validation on 5 random seeds.

Comparison Models	Data	MSE (↓)	MAE (↓)	PCC (↑)	SCC (↑)
TGDRP	COSMIC ^1^	1.9398 ± 0.0231	1.0435 ± 0.0058	0.8665 ± 0.0026	0.8164 ± 0.0074
DRPreter Template graph	COSMIC ^1^	1.9665 ± 0.0323	1.0435 ± 0.0089	0.8685 ± 0.0018	0.8232 ± 0.0022
DRPreter Template graph	Pathway ^2^	1.9276 ± 0.0495	1.0351 ± 0.0130	0.8711 ± 0.0034	0.8270 ± 0.0042
DRPreter w/o Trans ^3^ and Similarity	Pathway ^2^	1.8536 ± 0.0548	1.0085 ± 0.0123	**0.8820**±**0.0049**	**0.8445**±**0.0094**
DRPreter w/o similarity	Pathway ^2^	**1.8317**±**0.0276**	**1.0076**±**0.0067**	0.8778 ± 0.0018	0.8356 ± 0.0022

^1^ COSMIC: 702 COSMIC genes. ^2^ Pathway: 2369 genes of 34 cancer-related pathways. ^3^ Trans: transformer-based
cell-line–drug fusion module.

**Table 4 ijms-23-13919-t004:** Gradient-based gene importance analysis.

Drug	Cell Line	Disease	Top 5 Significant Genes	ln(IC50)	
				True	Predicted
Afatinib	GMS-10	Glioblastoma	*ACTR3B*, *PRR5*, *PRKCZ*, ***ERBB2***, *LTBR*	0.5372	0.5324
Vinblastine	NCI-H1792	NSCLC	*CYP7A1*, *GTF2H2*, *DVL2*, *RAB5B*, ***TP53***	−5.9258	−5.27633
Docetaxel	PANC0327	Pancreatic cancer	***CLDN18***, *SOX17*, *FGF19*, *WNT7A*, *CDH5*	−3.7668	−3.8204
Rapamycin	IGR1	Melanoma	*TYRP1*, *DCT*, *TYR*, *FRZB*, ***CDK2***	−1.6747	−1.7651
Bortezomib	EBC-1	Lung squamous cell carcinoma Derived from metastatic site: Skin	*SHC4*, *TNR*, *IL17RA*, ***MAPK12***, *SMURF1*	−5.7714	−6.0714

Genes in bold are direct targets of drugs, are involved in target pathways, or are biomarkers of disease.

**Table 5 ijms-23-13919-t005:** Notation of graph neural networks used in this paper.

Notation	Description
*G*	A graph.
*V*	Set of nodes of a graph.
*v*	A node included in V.
i,j	Indexes of the nodes.
*l*	Index of the layer of a graph.
$v_{i}$	*i*-th node in V.
$x_{i}$	Node feature of node $v_{i}$
*N*(*i*)	Set of neighbor nodes of a node $v_{i}$
*E*	Set of edges of a graph.
*A*	Adjacency matrix between nodes.
$W^{\left(l\right)}$	Trainable parameter matrix of *l*-th layer.
$X^{\left(l\right)}$	Node feature matrix of *l*-th layer.
σ	Nonlinear activation function softmax.
ϵ	Learnable parameter.

## Data Availability

The data and code used in this study are available at https://github.com/babaling/DRPreter, (accessed on 30 September 2022).

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
