# Peer review of "DRPreter: Interpretable Anticancer Drug Response Prediction Using Knowledge-Guided Graph Neural Networks and Transformer"

_ijms, 2022, doi:10.3390/ijms232213919_

Round 1

Reviewer 1 Report

Thank you for the opportunity to review your work. There are some minor revisions need. This paper constructed a drug response prediction implementing prior-knowledge and high explainability. My comments are as follows.

----comments----

#. Introduction: GOOD.

#. Figure 1: This is a very important figure to explain overall works of the present study. The figure quality and content delivery are very good. However, the explanation for this figure is very lacking. Each subsection in methods were not well matched with FIGURE 1.

#. Line 115: Difficult to understand. Is it correct?

- Each node v has node feature xi ∈ Rd , where d is a dimension of the i-th node feature.

- v >> i-th??

- It seems quite difficult to understand with expressions such as i-th and l-th, and mathematical expressions. Therfore, a schematic figure that describes how the data was formed and supplement the aforementioned expressions is recommended. 

#. Line 121: You introduced "message passing" as Key Mechanism!!. However, the theoretical background for it was not referred. 

- All following terms used in this paragraph were abstract: aggregate??, hidden state of each node??, k-hop neighbor nodes??, combine??

- What is output of AGGREGATE function?

- What is output of COMBINE function?

- A schematic figure is recommended to describe this paragraph.

#. Line 145 - How can you obtain 34 pathways, 2,369 genes, and 7,954 edges??, Were these obtained from KEGG?

#. Equation 2

- What is sigma??, sigmoid function??, activation function??

- What is W??

- Which figure matches this equation? If not, a schematic figure to describe this equation is recommended.

#. Equation 3

- What is ϵ??

- What is l??

- What is Xj??

- The content is not continuous and presented chaotically, making it difficult to understand.

- This is also a rationale to require the schmatic figure to describe equations 1 to 3.

#. Line 199: Why "l" 34??, Was it determined empirically?

#. Figure 2: lack of information

- How can you obtain raw drug embedding??

- How can you obtain raw drug pathway?? - which pathway used?

- What is final output? If Cell line-aware updated drug embedding is final outcome, detail description for it might be needed.

#. Line 247: transcriptomic data

- Did expression of transcriptomic data measure with RNA-seq or did with microarray?

- Are the transcripts summarised as ENSMBL ID, SYMBOL, or ENTREZ??

- What methods for scaling or normalization??

- Lack of information for transcriptomic data.

#. Line 250: protein-protein interaction

- No description or very few description in method.

- A detailed explanation or schematic figure about how KEGG and PPI are combined is needed.

#. Table 3: No pathways?

- It is recommended not only to present individual genes, but also to present important biological functions or pathways.

#. Figure 4: too lack of information or legend.

Although described in Figure 2, the x, and y axes in Figure 4 might be required to explained.

#. It is recommended that you provide a theoretical background as to why performance is good or comparable.

Reviewer 2 Report

In this manuscript, the authors propose a knowledge-guided graph neural network for anticancer drug response prediction and demonstrate the advancedness and validity of the model by ablation experiments and comparison experiments; the case study (biological significance mining) part of the article is outstanding compared with similar studies and should be recognized. At the same time, the authors need to answer some important questions and add some relevant experiments.

1.    The github link given by the author (https://github.com/babaling/DRPreter) was found to be non-existent; a check of the author's github page revealed that he has not submitted any code recently. Therefore, we cannot judge the authenticity of the experimental metrics reported in the author's paper, and we request the author to submit the relevant code for verification as soon as possible.

2.    Reporting the average test results under ten random divisions on a single dataset is not sufficiently reliable; we expect the authors to report cross-validation results on the study dataset and performance metrics on a new independent test set.

3.    The statement "According to the existing studies that suggest deep learning models for drug response prediction" in line 85 of the manuscript is not strong enough and we would like the authors to add 1-2 traditional machine learning methods (e.g. Random Forest and SVM) as baseline models for comparative model analysis.

4.    In addition to graph neural networks, there are many other types of deep learning methods that are widely used for drug response prediction, such as Jia et al., Nat Commun (doi: s41467-021-21997-5). Therefore, the authors need to illustrate the superiority of graph methods over other deep learning methods or specifically emphasize that this paper focuses only on graph neural network-based methods for anticancer drug response prediction.

5.    The statement in lines 411-412 "performance comparison experiments showed DRPreter has enhanced predictive power than the state-of-the-art drug response prediction models", is not appropriate and should be limited in scope.

6.    For recently published graph neural network-based drug response prediction models (e.g., Wei et al, Bioinformatics [doi: btac574]), authors should include in their discussion the advantages or highlights of their models compared to these models.

7.    The presentation in Figure 3 is not intuitive, and it is suggested that Figures 3a and 3b be combined and that statistical tests be performed to give significance indicators for differences in the distribution of predicted and known concentration values.

Round 2

Reviewer 2 Report

The authors have addressed most of my concerns.